# Relating Molecular Dynamics Simulations to Functional Activity for Gly-Rich Membranolytic Helical Kiadin Peptides

**DOI:** 10.3390/pharmaceutics15051433

**Published:** 2023-05-08

**Authors:** Tomislav Rončević, Matko Maleš, Yogesh Sonavane, Filomena Guida, Sabrina Pacor, Alessandro Tossi, Larisa Zoranić

**Affiliations:** 1Department of Biology, Faculty of Science, University of Split, 21000 Split, Croatia; troncevic@pmfst.hr; 2Faculty of Maritime Studies, University of Split, 21000 Split, Croatia; mmales@pfst.hr; 3Department of Physics, Faculty of Science, University of Split, 21000 Split, Croatia; sonavane2000@gmail.com; 4Department of Life Sciences, University of Trieste, 34127 Trieste, Italy; milenaguida@gmail.com (F.G.); pacorsab@units.it (S.P.); atossi@units.it (A.T.)

**Keywords:** glycine substitutions, helical antimicrobial peptides, mode of action, peptide–membrane interaction

## Abstract

Kiadins are in silico designed peptides with a strong similarity to *di*PGLa-H, a tandem sequence of PGLa-H (KIAKVALKAL) and with single, double or quadruple glycine substitutions. They were found to show high variability in their activity and selectivity against Gram-negative and Gram-positive bacteria, as well as cytotoxicity against host cells, which are influenced by the number and placing of glycine residues along the sequence. The conformational flexibility introduced by these substitutions contributes differently peptide structuring and to their interactions with the model membranes, as observed by molecular dynamics simulations. We relate these results to experimentally determined data on the structure of kiadins and their interactions with liposomes having a phospholipid membrane composition similar to simulation membrane models, as well as to their antibacterial and cytotoxic activities, and also discuss the challenges in interpreting these multiscale experiments and understanding why the presence of glycine residues in the sequence affected the antibacterial potency and toxicity towards host cells in a different manner.

## 1. Introduction

Antimicrobial peptides (AMPs) are important components of the innate defence systems of almost all forms of life and, in plants and animals, provide a first line of defence against invading pathogens [1,2]. They are being heavily investigated for their potential as novel anti-infective drug leads, particularly in the context of increasing bacterial drug-resistance as one of the great challenges of our age [3,4]. Novel antibiotics with alternative mechanisms of action are urgently required, and it is widely recognized that AMPs have multifaceted anti-infective roles, with multiple directly bactericidal capacities, and sometimes modulate the host’s immune and/or healing systems [5]. They usually have broad activity spectra, covering both Gram-negative and Gram-positive bacteria, including multidrug-resistant strains [6,7] that can also extend to biofilms, which, by one estimate, cause about 80% of human microbial infections [8].

Although the mechanism of action of many AMPs has been extensively studied, they remain elusive to a significant extent [9]. They most often target the negatively charged surfaces of bacterial membranes, which they destabilize by forming pores or other types of lesions [10,11], although the precise mechanisms of action can be quite diverse and complex, enabling them to adapt to bacterial counter-responses [12,13]. Nonetheless, some features are generally important, if not necessarily sufficient, for their activity, namely an overall positive charge that favours binding to negatively charged bacterial membranes and the segregation of polar and hydrophobic residues in different regions of an amphipathic structure, which facilitates insertion into the hydrophobic membrane core. A third feature, structural flexibility, is also relevant, as it enables them to rapidly adapt their conformation to the different environments they encounter on their trajectory from the bulk solution through the bacterial surface layers and into the lipid bilayer [14,15]. In this respect, interactions with components of the extracellular medium and of the bacterial outer membrane and/or peptidoglycan layers can be as important in defining their potency and spectrum of activity as the lipid compositions of the bacterial membranes they eventually interact with and disrupt [16,17,18]. Finally, the tendency of the peptides to remain monomeric or to oligomerize also significantly affects their mode of action and potency [19]. Considering that all these features must be correctly balanced to enhance the antibacterial potency while, at the same time, limiting the toxicity to host cells, it is understandable that AMPs must have quite subtle structure–activity relationship (SAR) characteristics.

Drug development efforts based on AMPs frequently involve tuning structural features that contribute to bactericidal activity while attempting not to increase the host cell toxicity, but for the above-mentioned reasons, this is not straightforward [4]. Several rational modifications are required to obtain the necessary information on the specific structural aspects that influence the potency and selectivity with respect to any given structural class of AMPs, which can then be used to build a more general picture. In this respect, small linear peptides that adopt a helical active conformation are probably the most studied among membrane-active AMPs, given the relative ease of redesign, preparation and structural characterization, and are therefore the best understood.

One effective approach, which has the required level of generality, is to develop sets of molecular descriptors that link physicochemical and structural properties with different aspects of the biological activity in quantitative structure–activity relationship (QSAR) models [20,21,22]. Using this methodology, we designed a self-consistent set of simple, linear, alanine and lysine-rich peptides that we have named kiadins [23,24] with at least 50% similarity to each other but with a variable number of glycine substitutions. They showed quite variable activity against a set of Gram-positive and Gram-negative bacterial pathogens, including some resistant strains, as well as variable cyto/genotoxicity when tested on human circulating blood cells. Flow cytometry and atomic force microscopy experiments confirmed that they act through membrane damage, leading to its permeabilization, but with subtly different modes of action. Molecular dynamics (MD) simulations in the presence of different models of biological membranes provided some explanation, indicating that peptides’ affinities for these vary in a manner that is apparently influenced by the number and placement of glycine residues along the sequence [23,24,25].

In this paper, we expand the MD analyses to obtain more information on the underlying reasons for differences in interactions with both the neutral 1,2-dilauroyl-sn-glycero-3-phosphocholine (DLPC) bilayer (used as the model for the eukaryotic cell membrane) and with the negatively charged POPE:POPG (1-palmitoyl-2-oleoyl-sn-glycero-3-phosphoethanolamine:1,2-dipalmitoyl-sn-phosphatidylglycerol) bilayer (model for the bacterial cell membrane), observing structural changes and interactions that occurred during a simulation time of 300 ns. We then related these observations to structural changes observed in the presence of liposomes with similar phospholipid compositions as in the simulations using circular dichroism (CD) spectroscopy, as well as differences in the interactions with immobilized liposomes using surface plasmon resonance (SPR). Finally, we attempted to relate the computationally and experimentally determined observations on structure variations and membrane interactions with the antibacterial potency, which had been previously determined against two reference bacteria (Gram-negative *Escherichia coli* and Gram-positive *Staphylococcus aureus*) [23,24], and cytotoxicity towards human red blood cells (RBC) and the MEC-1 lymphocyte-derived cell line. In particular, we attempted to understand the reasons underlying the fact that the antibacterial potency or toxicity towards host cells are affected quite differently by the proportion and position of Gly residues in the sequence. This is an important aspect in view of developing AMPs for therapeutic uses.

## 2. Materials and Methods

### 2.1. Peptide Design and Synthesis

Kiadin-2 and -3 were previously designed by modifying a parent peptide, kiadin-1, itself derived from the natural AMP (PGLa-H) [24], as described in detail in [23]. Kiadin-4, -5 and -6 were ab initio designed by following a defined set of rules based on observed features from a set of natural peptides, as also described by Roncevic et al. [23]. Briefly, these rules required the presence of recurring motifs that are often observed in helical AMPs, as well as a threshold amphipathicity, allowing a certain structural flexibility so that the helices could be distorted towards a 3_10_ conformation. Furthermore, the sequence should result in a D-descriptor value (a previously described QSAR selectivity descriptor based on hydrophobicity) [21,26] that predicts a high selectivity index (SI) based on a training set of helical anuran peptides with known antimicrobial and cytotoxic activities [21,27], as implemented by the Mutator QSAR tool for AMP design. Irrespective of the design procedure, the peptide sequences are sufficiently similar (≥50% identical residues among them) to be considered as a single family named *kiadins*.

C-terminally amidated peptides were obtained from GenicBio Limited (Shanghai, China) at >98% purity, as confirmed by RP-HPLC and MS (see Appendix A). Peptide stock concentrations were determined by dissolving accurately weighed aliquots of peptide in doubly distilled water and further verified by using the extinction coefficients at 214 nm, calculated as described by Kuipers and Gruppen [28].

### 2.2. MD Simulations

Molecular dynamics simulations were carried out on *di*PGLa-H (tandem repeat) and the six designed kiadin peptides (see Table 1) in a closed surrounding consisting of a neutral DLPC bilayer or a negatively charged POPE:POPG bilayer immersed in water, as described previously [23]. Briefly, the initial coordinates for all peptides were furnished by the QUARK template-free protein structure predictor [29,30], which returned α-helical structures for all sequences. Gromacs version 2019.3 [31] was used for all simulations, applying the GROMOS (54a7) force field [32,33,34] to peptide and DLPC lipid molecules and the GROMOS-CKP force field to POPE and POPG lipid molecules [35,36,37], while the SPC (simple point charge) model was used for water molecules [38].

Two simulations were carried out for each peptide with each membrane type. In *case1*, the peptide molecule was positioned facing its more polar surface and, in *case2*, its more hydrophobic surface towards the membrane (snapshots of the initial conformations are shown in Figure 1 and Appendix A with more detailed descriptions of the peptide positions).

The neutral lipid bilayer was created by placing 128 DLPC molecules in each leaflet and was solvated (keeping the water–lipid ratio to >50) as described previously [23]. The anionic system was created by placing 96 POPE and 32 POPG lipids in each leaflet (3:1 mixing ratio, with an equal number of L- and D-configurations [39]), and the water–lipid ratio was 40. The POPE and POPG lipid configurations were obtained from the Lipidbook repository [40], while the MemGen online tool [41] was used to generate the anionic membrane.

The peptide charge was defined for pH 7 (see Table 1), considering a free N-terminal amine but amidated C-terminus. Cl^−^ counter ions were added to neutralize the peptide charges, while Na^+^ ions were also added to neutralize the negatively charged POPE:POPG bilayer. Details of the equilibration process are given in the Appendix A. Production runs of 300 ns, sufficient to observe the effects of peptides binding to the membrane surface, were performed in isothermal–isobaric (constant NpT) ensemble T = 310 K and *p* = 1 bar (using the Nosé–Hoover thermostat and Parrinello–Rahman barostat, with a 0.5 ps time constant for the temperature and 2.0 ps for the pressure, and compressibility equal to 4.5·10^−5^ bar) [42,43]. The leapfrog integrator time step was fixed at 2 fs, and the bonds were handled by the LINCS option [44]. The particle mesh Ewald method [45] was used for the calculation of the electrostatic interaction, while the van der Waals cut-off was set to 1.0 nm. The simulation time was 7 × 2 × 300 ns = 4.2 μs for both DLPC and POPE:POPG systems, which resulted in an overall simulation time of 4.2 × 2 = 8.2 μs.

Post-run analyses were performed mainly by using Gromacs utilities, such as the *gmx* routine *mindist*, which computes the number of contacts and minimum distance between one group and a number of other groups, and *gmx density* utility, which computes density profiles [31]. The time dependence of the secondary structure was calculated using the DSSP program [46]. The Eisenberg hydrophobic moments were measured using HeliQuest [47]. The 3D hydrophobic moment, based on the actual molecular structure for the initial and final conformations (collected after 300 ns of simulation time) was determined using the online server 3D-HM (3D Hydrophobic Moment Vector Calculator) [48]. In this case, the HM vector is evaluated from the surface charge distribution of the hydrophobic and polar regions over a given shape, so that it is not limited to a perfect helical structure, as is the case of the Eisenberg (2D) hydrophobic moment provided by HeliQuest. The figures were prepared using VMD [49].

### 2.3. Preparation of Liposomes

Large unilamellar vesicles (LUVs) were prepared as described by Morgera et al. [50]. Briefly, dry 1,2-dioleoyl-sn-glycero-3-phosphocholine (DOPC) or phosphatidylglycerol:diphosphatidylglycerol (DPPG) was dissolved in chloroform/methanol (2:1) solution, then evaporated using a dry nitrogen stream and vacuum-dried for 24 h. The liposome cake was resuspended in 1 mL of sodium phosphate buffer (SPB) to a concentration of 5 mM phospholipid and spun for 1 h at 40 °C. The vesicles were then subjected to several freeze–thaw cycles before passing through a mini-extruder (Avanti Polar Lipids, Alabaster, AL, USA) with successive polycarbonate filters having 1, 0.4 and 0.1 μm pores before resuspending to a final phospholipid concentration of 0.4 mM. By calculating the surface area of a ∼100 nm liposome and taking into account the average area of a phospholipid headgroup (∼0.7 nm^2^) [51] (similar to the value determined in MD simulations), the concentration of the liposomes was estimated to be about 5 nM.

### 2.4. Circular Dichroism

CD spectra were obtained on a J-710 spectropolarimeter (Jasco, Tokyo, Japan) as the accumulation of three scans. They were measured in the presence of DPPG and DOPC LUVs, prepared as described above, suspended in 10 mM sodium phosphate buffer (SPB) using a 2 mm path length cuvette, 20 μM peptide and a 190–240 nm wavelength range. The % helix content was determined as [θ]^222^/[θ]^α^, where [θ]^222^ is the measured molar per residue ellipticity at 222 nm under any given condition and [θ]^α^ is the molar ellipticity for a perfectly formed α helix of the same length, estimated as described by Chen et al. [52].

### 2.5. Antimicrobial Activity

The antibacterial activity of the kiadins was previously reported by Roncevic et al. [23] and was assessed using the microdilution method in 96-well microtiter plates according to EUCAST guidelines [53]. Overnight grown bacteria, *Escherichia coli* ATCC 25922 or *Staphylococcus aureus* ATCC 25923 reference strains, were cultured in fresh Mueller–Hinton broth and were added to a serial dilution of peptides at a concentration of 5 × 10^5^ CFU/mL in 100 μL per well. The minimum inhibitory concentration (MIC) was taken as the lowest concentration of the peptide showing no visual bacterial growth after an overnight incubation at 37 °C. The minimum bactericidal concentration (MBC) was determined as the lowest concentration with no visible bacterial growth on MH agar after 18 h of incubation at 37 °C of peptide–bacteria aliquots taken from wells corresponding to the MIC, 2 × MIC and 4 × MIC.

### 2.6. Toxicity Assays

Haemolytic activity was previously reported by Roncevic et al. [23] and was evaluated by incubating 0.5% erythrocytes provided by an informed healthy donor with serially diluted peptides in 96-well microtiter plates. Haemoglobin release was monitored at 450 nm. Triton-X was used as a positive control, and the incubation of red blood cells in PBS without peptides was the negative control. A HC_50_ value was calculated as the concentration causing 50% haemolysis.

Cytotoxicity was also assessed on a human lymphocytic cell line by using the 3-(4,5-dimethylthiazol-2-yl)-2,5-diphenyl tetrazolium bromide (MTT) assay. The MEC-1 chronic lymphocytic leukaemia [54] cell line was kindly provided by Prof. P. Macor (Department of Life Sciences, University of Trieste, IT) [55]. MEC-1 cells were cultured in RPMI 1640 complete medium supplemented with 2 mM L-glutamine, 100 U/mL penicillin, 100 µg/mL streptomycin and 10% foetal bovine serum (FBS) and were subcultured two/three times a week for not more than 20 passages. The colorimetric MTT assay was performed to assess the metabolic activity of cells plated onto 96-well culture plates (10^5^ cells/well, V^f^ = 100 µL) that had been exposed to 5 to 50 µM kiadins for 24 h. Twenty microlitres of a stock MTT solution (5 mg/mL) were added to each well after 20 h and left to incubate for the remaining 4 h. The converted MTT dye was then solubilized with acidic isopropanol (0.04 N HCl in absolute isopropanol). Absorbance was measured at 540 and 630 nm using an automated microplate reader (EL311, BIO-TEK^®^ Instruments, Charlotte, VT, USA). All measurements were carried out at least in triplicate. An IC_50_ value was estimated as the concentration of peptide causing a 50% decrease in viability, calculated using GraphPad Prism software.

### 2.7. Surface Plasmon Resonance

Peptide–membrane interactions were studied using an X-100 instrument (Biacore, GE Lifesciences). LUVs were prepared using DOPC as described above. The extruded unilamellar LUVs were then diluted to result in ~1 mM phospholipid in PBS. The L1 sensor chip was preconditioned and loaded with the LUVs, as described previously [56]. After loading, 0.1 mg/mL BSA was injected to cover any exposed nonspecific binding sites for peptides. Only a very small increase in the signal was observed (~60 RU), confirming that coverage of the surface with liposomes was effectively complete. This procedure results in integral liposomes on the chip surface without fusion [57]. To detect the binding of peptides with the liposomes, solutions with increasing peptide concentrations (0.5 to 16 μM) were injected sequentially, at a constant flow rate of 10 μL/min, for a contact time of 540 s. This was followed by a dissociation time of 1200 s by washing with a PBS buffer to allow the peptides to return into the bulk solution. Sensorgrams for each concentration were collected using BIAevaluation software v 1.1. At the end of each run, the flow cell was regenerated by flowing isopropanol and then NaOH solution to remove all phospholipids before being treated with fresh LUV suspension (only three regenerations at most were used). Each experiment was repeated three times with a given peptide, resulting in very similar curves. The best fit for the binding curves was obtained with the “affinity-steady state” mathematical model in BiaEvaluation software, from which it was possible to obtain an equilibrium dissociation constant (K_D_).

## 3. Results

### 3.1. Kiadin Peptides

Kiadins are a series of short, linear peptides that were previously designed based either on the sequence of PGLa-H (KIAKVALKAL) used in a tandem to produce *di*PGLa-H and then subjected to a single or to a double V→G substitution (kiadins 1–3, see Table 1) or designed ab initio according to a predefined protocol, as described previously (kiadins 4–6) [23]. The two sets of peptides show different patterns of hydrophobic and charged residues in some positions but have at least 50% of the residues in common with *di*PGLa-H, so they can be considered to form a self-consistent series of seven peptides with the same length (20 residues), C-terminal amidation and charge (+7). They sample three different single Gly substitutions (V^5^→G, V^15^→G and K^14^→G); one double Gly substitution (V^5^→G + V^15^→G) and two quadruple Gly substitutions (A^3^→G + L^7^→G + K^14^→G + K^18^→G and K^4^→G + L^7^→G + K^14^→G + L^17^→G).

Gly residues are neither polar nor hydrophobic and introduce flexibility to peptide conformations. Substitutions were carried out in such a way that the overall charge was not altered, and although the overall hydrophobicity decreases somewhat with an increasing number of Gly residues, the amphipathicity (measured in terms of the Eisenberg 2D hydrophobic moment) is affected differently, so that kiadin-4 and -5, each with four Gly residues, respectively, have the highest and lowest 2D-hydrophobic moments (2D-HM) calculated using HeliQuest [47] (see Table 1). This, however, is an approximate measure of amphipathicity based on helical wheel projections (see Appendix A) that assume a perfectly helical structure. The 3D-hydrophobic moment [48] calculated using atomic point charges and based on the molecular structure is likely a more realistic estimate of amphipathicity and also shows the lowest 3D-HM for kiadin-5 for the initial structure predicted by the Quark server [29,30] (see Table 1). During the MD simulation time, the 3D-HM changes significantly for all peptides (see Table 1), as also shown representatively in Figure 2 for kiadin-2 interacting with the anionic membrane (*case1* orientation). Visual representations of the initial 3D-HM for all kiadins are shown in Appendix A, and the effect of their interactions with both types of membranes (for both initial orientations or *cases*) are shown in Appendix A, respectively.

### 3.2. Peptide–Membrane Interaction Simulations

Molecular dynamics runs were carried out to simulate the initial stages of the interaction of kiadins with the surface of a membrane formed by either the neutral or anionic model membranes [39]. This would subsequently lead to a deeper membrane insertion, and more or less efficient lytic mechanisms and simulations provide a microscopic insight that can inform on the possible ensuing mechanism/s of action, if used with caution [18,58,59]. We therefore subsequently attempted to determine if these observations could be correlated with experimental data collected for the peptides interacting with real membranes. This is, at best, tentative, given the different starting conditions used and the quite different time scales that are covered (ns to hours), but any correlation could be quite revealing.

We applied a moderately long simulation time in two independent runs with different sides of the amphipathic peptide initially facing the membrane surface (polar side for *case1* and hydrophobic side for *case2*) to more completely assess its interaction with the planar-solvated membrane surface. We examined different aspects such as:

(I) Peptide interactions and contacts. By monitoring both the total number of contacts between peptide atoms and membrane P atoms (averaged over the final 100 ns of the simulation) and the number of contacts between Lys side-chain atoms (polar sector) and membrane P atoms (again, averaged from the last 100 ns of the simulation time), we found, as expected, that the cationic kiadins tended to bind more rapidly to, and form more contacts with, the anionic POPE:POPG membrane than with the neutral DLPC membrane (compare to Appendix A). However, there are some exceptions (see Appendix A), with *di*PGLa-H and kiadin-1 and -5 forming more contacts with the neutral membrane in some simulations. Contacts formed by the polar side of the peptides (Lys side-chain atoms with membrane phosphate atoms) are also informative (see Appendix A). There is a decided preference for the peptides to interact more with their N-terminal region, particularly with respect to the anionic membrane. For neutral membranes, there are some notable exceptions, such as kiadin-2.

(II) Effect on peptide structure. Kiadins show quite different helix stabilities, as determined using the DSSP program, over the 300 ns simulation time (see Appendix A), which, however, did not necessarily correlate with the number of Gly residues present in the sequence. Kiadin-2, -3 and -6, with one or two Gly residues in different positions, showed more stable structures overall than the parent peptide *di*PGLa-H, with no Gly residues. The DSSP plots indicated that conformational distortion occurs, with a general loss of helicity and the significant presence of π-type helicity causing bulges in the helical conformation in some cases. Even four glycine residue substitutions do not necessarily abolish the conformational stability, as shown by kiadin-4. Considering the average helical content over the 300 ns simulation time (Appendix A), *di*PGLa-H and its monosubstituted analogue kiadin-1 were, in fact, the least stable overall, while the monosubstituted analogues kiadin-2 and -6 were the most stable. The tetra-substituted kiadin-4 and -5 showed a marked dependence of their stability on the type of membrane with which they interact.

(III) Role of peptide amphipathicity. The hydrophobic moment calculated based on the molecular structure (3D-HM) changes for all peptides in membrane interactions, with the vector magnitude often increasing significantly (see Table 1 and Appendix A, as well as Appendix A). This takes into account the fourfold decrease in the dielectric constant on passing from an aqueous bulk solution (D_e_ = 78.54) to the less polar membrane surface region (estimated at D_e_ ~ 20 [42] and suggests that the peptides may adapt their structure to increase the amphipathicity in membrane interactions. In most cases, the orientation of the 3D-HM vector at the end of the simulation is directed along the membrane surface, tilting away from membrane bilayer, i.e., the polar side of a peptide is more in contact with the membrane surface, which creates a favourable electrostatic dipole for the initial interaction. This is particularly the case for the anionic membrane (φ angle in general ~50° to ~85°). Changes in the 3D-HM vector angle are more varied in interactions with the neutral membrane (φ angle ~30° to ~170°). The vector magnitude is, on average, higher when an interaction occurs with the anionic membrane (average 3D-HM ~45 kTÅ/e vs. ~33 kTÅ/e) but with significant variations, depending on the peptide and, also for some peptides, its initial orientation (*case1*).

(IV) Peptide positioning and binding. Surface binding of the kiadins to neutral or anionic membranes can be represented using density profiles (see Figure 3, Figure 4, Figure 5 and Figure 6) that compare the distribution of peptide atoms (black curve) or of lysine side-chain atoms (yellow curve) relative to the bilayer surface (represented by the distribution of P atoms, blue curve). As already indicated, *di*PGLa-H loses α-helicity with time and positions mainly on the surface of the membrane during the simulation. It only partially retains its structure in the middle region when interacting with the hydrophobic side of the anionic membrane (*case2;* see Figure 6 and Appendix A). Kiadin-1, with a single Gly substitution in the C-terminal region, shows a similar behaviour, and destabilization of the helical conformation is evident in most cases. When interacting, from the hydrophobic face with the anionic membrane, it adopts a π-helical structure (53%); with the neutral membrane, it adopts a compact structure composed of β-strands (49%). The peptide enters deeply into the anionic membrane polar region, as seen in the density profiles.

Kiadin-2, with a single Gly substitution in the N-terminal region, shows a higher conformational stability that is less dependent on the membrane type and approach orientation, maintaining a >40% α-helix content, on average, for all simulations (Appendix A). This does not, however, result in more contacts with membrane phosphate atoms, and unlike most of the other peptides, it does not interact with the DLPC membrane phosphates preferentially with its N-terminal Lys residues but reorients markedly, and although it does not insert deeply into the membrane surface, this may be a first step towards a subsequent insertion. Conversely, when interacting with the anionic membrane, the peptide enters quite deeply into the surface in a less tilted manner. A visual inspection (see the final snapshot in Figure 3, Figure 4, Figure 5 and Figure 6), as well as the variation in 3D-HM, which increases and, in most cases, orients towards the membrane surface, suggest a possible two-step binding mechanism. The N-terminal segment unfolds and enters onto the surface, pulling the peptide into the polar region and flipping it to present the hydrophobic side toward the bilayer and leaving the polar sector in contact with the negative membrane charges. Kiadin-6 also has a single Gly substitution in a position close to that of kiadin-1 but behaves more similar to kiadin-2, maintaining an over 30% helicity in all simulations, although it is quite distorted when binding to both types of membranes. The variation in the 3D-HM suggests a similar change to kiadin-2 in orientation once the peptide starts entering the DLPC membrane.

Kiadin-3, with two Gly substitutions, retains some helical stability in all cases that is less than kiadin-2 but more than kiadin-1. When interacting with the neutral membrane, the conformational change depends markedly on the initial orientation, so that it exhibits a destabilization of the central region for *case1*, similar to kiadin-6, and a transition to a β-hairpin for *case2*, similar to kiadin-1.

Kiadin-4, which has four Gly substitutions, retains a greater conformational stability when interacting with the neutral DLPC membrane than on contact with the anionic POPE:POPG membrane, where it effectively completely unfolds. It does not seem to insert deeply into either membrane. Conversely, kiadin-5, with a different pattern of 4 Gly substitutions, unfolds completely when binding to the DLPC membrane, while it preserves some α-helical structure in contact with the anionic membrane. This notwithstanding, it does not insert deeply into either membrane.

### 3.3. Kiadin Structural Stability Determined by Circular Dichroism Spectroscopy

An attempt was made to relate the structural variations observed during MD simulations with the experimentally determined conformational behaviour as assessed by CD spectroscopy in the presence of the neutral or anionic liposomes (Figure 7). This, however, probes a much later stage of the interaction with the membrane and samples both membrane-bound peptides and unbound peptide populations. CD spectra in the presence of the neutral LUVs indicate that most kiadins either do not interact to a significant extent or do not adopt a helical structure when they do so, as only a slight conformational change is observed with respect to the aqueous solution (<10% helix), the only exception being kiadin-6 (~20% helix). Neutral LUVs are considered to model the membranes of eukaryotic cells so that the conformational changes observed by CD in their presence are often predicted to correlate with the cytotoxicity.

Conversely, a marked conformational transition is observed by CD in the presence of anionic LUVs (see Figure 7). This suggests that a strong interaction occurs with the anionic membrane in a manner that results in a bound population of peptides with a well-defined conformation. However, with the exception of kiadin-5, all the spectra deviate significantly from those expected for a canonical α-helical conformation, which shows distinctive minima centred at 208 and 222 nm and a θ_222_:θ_208_ ratio ≥ 1. Possible explanations for the altered spectra could be the formation of other types of helical conformations (e.g., 3_10_ or π-helix) or aggregation of helices to form bundles, which has been reported to invert the θ_222_:θ_208_ ratio [23].

It is therefore possible to estimate the α-helix content from the CD spectrum only for kiadin-5 (~50%), while, for the other peptides, one can only surmise that kiadin-2 and -3 interact with the anionic membrane in a manner leading to a more stable conformation than for kiadin-1, -4 or -6 and that this conformation deviates significantly from that of a canonical α-helix. Curiously, *diPGLa*-H, with no Gly substitutions, has the weakest interaction. MD simulations in the presence of the anionic membrane, albeit sampling only the initial stages of membrane interactions, are consistent with the peptides adopting conformations that deviate somewhat from the canonical helical one. With respect to the neutral membrane, MD simulations suggest that peptides generally do not interact strongly, remaining on its surface.

There are two possible explanations for our CD observations, taking into account that the peptides are partitioned between two environments, a bulk solution and the membrane bilayer. One is that, while the majority of peptide molecules interact with the anionic membrane in a manner that alters their conformation, only a small minority of peptide molecules partition into neutral membranes, so that the conformational effects occurring in these molecules cannot be discriminated. Another is that they interact with both membranes, but only anionic ones induce a marked degree of structuring. The interactions of the peptides with neutral membranes therefore needed to be probed using alternative methods, such as surface plasmon resonance, that respond only to the bound peptide population.

### 3.4. Binding to Neutral LUVs Determined Using Surface Plasmon Resonance

*Di*PGLa-H shows the typical binding sensorgram for a peptide that binds rapidly to, and eventually saturates, the membrane surface (see Figure 8 and its inset). From the binding curve, it was possible to estimate a K_D_ of ~5 µM. Kiadin-1, with a C-terminal Val^15^→Gly substitution, binds with a comparable affinity up to 4 μM, but then, there is a jump in the response unit (RU) signal (Figure 8), suggesting a second type of interaction possibly involving the aggregation of peptides on the membrane surface. This results in a linear correlation of the signal with the peptide concentration (see inset), so it is not possible to estimate a binding constant, as saturation does not occur in the concentration range used. It also appears to result in slower release kinetics from the membrane when washing with a buffer. Kiadin-2, with a N-terminal Val^5^→Gly substitution, binds with a similar affinity to *di*PGLa-H up to a peptide concentration of 8 μM (Figure 8). However, at 16 μM, after an initial increase in the response, there is a sharp drop in the RU, a behaviour we have previously associated with an AMP-induced LUV breakup [19]. Kiadin-6, which also has a single substitution (Val^14^→Gly), shows the strongest increase in the RU for peptide binding (Figure 8), suggesting a higher affinity for neutral LUVs, consistent with CD observations. The behaviour of the sensorgrams and binding curve indicate that, similar to kiadin-1, self-aggregation occurs at the membrane surface, starting at a low threshold concentration. Thus, the singly substituted kiadins show a spectrum of binding behaviours, ranging from binding until saturation to surface aggregation or to membrane disruption, even after relatively limited modifications in the sequence.

Kiadin-3, with the double Val^5,15^→Gly substitution, behaves more similar to *di*PGLa-H than either kiadin-1 or -2, showing no surface aggregation and having a similar binding constant (Figure 8). Kiadin-4, with a quadruple Gly substitution (Figure 8), maintains a similar affinity for the membrane but shows a significant degree of irreversible binding at the highest concentration used, possibly indicating translocation into the LUVs. Kiadin-5 also has a quadruple Gly substitution but with a different residue pattern to kiadin-4 and shows a markedly lower affinity for neutral LUVs (Figure 8). The shape of the sensorgrams suggests a two-stage binding process, which is far from saturation. This strongly suggests a different and significantly weaker type of interaction with the neutral membrane than the other peptides, which is consistent with the CD results.

Taken together, the SPR results confirm that the peptides present a range of behaviours with respect to binding to the DOPC LUVs, with the parent peptide *di*PGLa-H and kiadin-3 interacting with the membrane surface in a manner that reaches saturation. Kiadin-2 behaves similarly but, at the highest concentration used, starts disrupting the LUV integrity. Kiadin-6 aggregates strongly on the membrane surface in a manner that does not reach saturation, suggesting surface aggregation, with kiadin-1 and -4 showing intermediate behaviours. Kiadin-5 shows the lowest binding capacity of all the peptides. Furthermore, for some peptides, partial irreversibility of the signal may indicate that some translocation into the LUVs may occur.

### 3.5. Antibacterial Activity

The potency of *di*PGLa-H and kiadin peptides has been reported previously by Roncevic et al. [23] on a panel of Gram-negative and Gram-positive bacteria. Selecting *E. coli* and *S. aureus* as representative reference strains, significant differences in activity are evident. The precursor tandem repeat peptide *di*PGLa-H shows quite potent activity on both types of bacteria, with sub- to low micromolar MIC and MBC. Kiadin-1 (Val^15^→Gly) gains activity against *E. coli*, with submicromolar MIC/MBC, while maintaining its activity against *S. aureus*. Interestingly, kiadin-2 (Val^5^→Gly) significantly increases the activity against *E. coli* but loses it markedly with respect to *S. aureus* (see Table 2). This suggests that Gly-induced conformational changes may not only affect how the peptide interacts with membranes but also how it approaches through the quite different outer layers of Gram-positive (thick peptidoglycan) and Gram-negative bacteria (outer membrane with lipopolysaccharide). Kiadin-6 (Val^14^→Gly) instead shows a somewhat decreased potency against both bacteria. Thus, mirroring what occurs in the SPR, relatively small variations in the sequence result in significant variations in the effect—in this case, on the antimicrobial activity and spectrum.

The introduction of a double substitution in kiadin-3 or a quadruple Gly substitution in kiadin-4 more markedly reduces the activity against both species but less so for *E. coli*. Introducing a different pattern of quadruple Gly substitution completely abrogates the activity in kiadin-5. Thus, the presence of Gly substitutions in the kiadins series of peptides has a significant effect on the antimicrobial potency and specificity but depends markedly on the sequence context of these substitutions.

### 3.6. Toxicity

Different aspects of the cytotoxic effect of kiadins on human cells were probed using a haemolytic activity assay of RBC membrane integrity [23] and the MTT colorimetric assay of cell mitochondrial viability using the MEC-1 B-chronic lymphocytic leukaemia cell line (see Figure 9). With respect to the haemolytic activity, under the conditions used, the parent peptide *di*PGLA-H was moderately cytotoxic, with a HC_50_ estimated at about 22 μM (Table 2). Kiadin-1 and -2 show a similar behaviour with comparable HC_50_ values, whereas kiadin-6 is significantly more cytotoxic. Kiadin-3 has a threefold higher HC_50_ estimated at about 95 μM. This does not, however, relate to the increased number of Gly residues, as kiadin-4 (4 Gly) shows significant haemolytic activity, with a HC_50_ comparable to that of kiadin-6. On the other hand, kiadin-5 (also with 4 Gly) appears not to be haemolytic up to the highest tested concentration.

Cell viability correlates moderately well with haemolytic activity, except that *di*PGLa-H was found to be more cytotoxic to MEC-1 cells, with an IC_50_ of 9 ± 3 μM. Again, kiadin-3 was significantly less cytotoxic than the other peptides, with IC_50_ values above the tested range (>50 μM). Taken together, there is also some correlation between the cytotoxic activity, as determined in these experiments, and binding to the neutral liposomes, as determined by the CD and SPR experiments. Kiadin-6, which shows significant binding by both of these methods, is also quite cytotoxic, suggesting that its capacity to bind to this type of membrane plays a role in its cytotoxic activity. Furthermore, MD simulations appear to show the beginning phase of its insertion into this type of membrane. Kiadin-3 is significantly less cytotoxic, does not bind as efficiently in the SPR and CD experiments (see Figure 8) and is less inserted in the MD simulation.

Kiadin-5, which binds least with the neutral LUVs (as determined both by the CD and SPR) and, in the MD simulation, completely loses its helical conformation on the membrane surface, is also the least cytotoxic. For the other kiadins, relating the binding behaviour to the cytotoxicity is complicated by the fact that they apparently aggregate on the membrane surface, which may affect their subsequent mode of action.

## 4. Discussion

Kiadins were designed either by modifying a parent peptide, *di*PGLa-H, using a QSAR-based program or converging onto this sequence when using a rule-based ab initio design process [23]. They provide a self-consistent set of sequences for investigating the effect of Gly substitution on the structural characteristics of helical AMPs and how this affects their membrane-related mechanism of action, modulating the antimicrobial activity and toxicity.

The MD simulations indicate that all the kiadins interact rapidly with both the anionic and neutral membranes, altering their conformations in the process—sometimes unfolding completely or else adopting distorted helical conformations or even other types of conformations, a process that seems to be highly dependent on the type of membrane. This early stage of interaction with the membrane surface is consistent with more extended conformations, which may gradually adopt a helical content to assist in peptide insertion, but other types of conformations with favourable 3D hydrophobic moments may be sampled as well. With respect to the type of Gly substitution, taken together, the results of the simulations indicate that their effect on the structural stability depends markedly on their position in the sequence and the type of membrane they interact with (neutral or anionic) but not necessarily on the number of substitutions.

The CD studies show they all interact with anionic membranes at a later stage in a manner that favours a conformational change, though, in most cases, not to a canonical α-helix. In fact, kiadin-5 is the only one that does show a canonical helical conformation, consistent with the MD results, but is the least active on bacteria. This suggests that interaction with the anionic membrane, leading to helical structuring, is, by itself, insufficient for antimicrobial activity but needs to be followed by subsequent effects. For example, peptide aggregation does result in a deviation from a canonical helical structure (as is observed in the CD spectra of the other peptides) and, to some extent, is confirmed for some kiadins by the SPR experiments.

With respect to neutral liposomes, the CD spectra indicate a significant helical content only for kiadin-6, which would suggest that kiadins do not interact strongly with this type of membrane in general, although some are quite cytotoxic to circulating blood cells. It should, however, be considered that CD spectra are a composite result of all conformations present in the peptide population, so that either kiadins insert into this type of membrane in a manner that does not affect their conformation, which is improbable, or the binding equilibrium strongly favours the peptides remaining in the bulk solution or on the surface of the membrane, remaining unstructured. Their contribution to the CD spectrum would drown the small proportion of peptides that do insert into the membrane bilayer and adopt a more regular conformation. Only for kiadin-6 does an appreciable proportion of the peptide population seem to do this. Thus, cytotoxicity may derive from the relatively low proportion of molecules that do insert into eukaryotic-type neutral membranes (estimated to be <10% for most kiadins from the CD spectra). One should therefore be cautious in interpreting CD spectra in the presence of neutral LUVs to predict cytotoxicity or the lack of it for AMPs.

Our CD experiments therefore suggest that, for kiadin-like peptides, at least, (i) the observation of a well-defined, canonical helical structure for a peptide in the presence of the anionic membranes is not necessarily an indication that it will be a good AMP, (ii) deviation from this to a noncanonical helix-type conformation may instead correlate with increased antimicrobial activity and (iii) increased helical conformation in the presence of neutral membranes may correlate with increased cytotoxicity but a lack of it does not necessarily correlate with low cytotoxicity. Experiments with other types of helical peptides are required to reveal if this may be a more general rule, and in any case, it is appropriate to complement CD with other types of measurements, such as SPR, which probe only the membrane-binding population and can discriminate different types of binding.

The SPR studies did, in fact, confirm that all peptides showed rapid binding kinetics with the neutral LUVs, suggesting good adhesion capacity to the membrane surface, although the sensorgrams had significantly different concentration-dependent profiles, suggesting different interaction/insertion mechanisms. Some kiadins appeared to aggregate in the membrane surface, others to accumulate without aggregating in a manner that causes membrane disruption. In any case, these studies indicated that kiadin-5 has the least binding/insertion capacity and kiadin-6 the most, which are consistent with their CD behaviour and also correlates with their being the least and most cytotoxic toward host cells. A difference in the binding mechanism is, to some extent, also observed in the MD simulations, where one Gly mutation near the C-terminus of kiadin-1 introduces high flexibility in the structure that could promote peptide self-aggregation, as observed with SPR, while, in kiadin-2, which has Gly near the N-termini, it enables the peptide to form a kinked helix that may facilitate peptide insertion and membrane rupture without aggregation at higher peptide surface concentrations.

CD and SPR are often used to predict the cytotoxicity of AMPs, which, in our case, was assessed experimentally using haemolysis and MTT assays. These two assays look at different aspects of a peptide’s interactions with host cells that are far more complex than interactions with the naked model membranes of neutral liposomes. The former looks at the capacity of the peptides to damage the cytoplasmic membrane, whereas the latter probes the capacity of peptides to affect the mitochondrial activity, which is a subsequent effect to membrane damage or transit. Nonetheless, there is an overall reasonable correlation with the results from the simpler biophysical studies (i.e., CD and SPR), especially regarding kiadin-5 and -6.

Possibly the most interesting aspect of the effect of Gly-substitutions on the biological activities of kiadins is the markedly different effects they can have on antimicrobial potency with respect to host cell cytotoxic activities. We have previously reported how the presence and placement of Gly substitutions in designed helical AMP sequences can markedly affect their antimicrobial potency and selectivity [60,61]. In particular, we underlined that it is more the placement than the number of substitutions that affects the potency and that, in general, it affects the activity towards Gram-positive bacteria and cytotoxicity more than the activity on Gram-negative microorganisms. The present work confirms these previous observations to some extent, although it shows how complex the relationship can be. Thus, a single substitution in kiadin-1 slightly benefits the antimicrobial potency without markedly affecting the antimicrobial selectivity or cytotoxicity, while, at a different position in kiadin-2, it benefits the potency against the Gram-negative bacterium and reduces the cytotoxicity but at the expense of a decreased potency against the Gram-positive bacterium. For kiadin-6, the single substitution not only decreases the overall antimicrobial potency but it also markedly increases the cytotoxicity, even more so for the tetra-substituted kiadin-4.

## 5. Conclusions

Our study of kiadins confirms previous observations [60] that the position of Gly residues is of the utmost functional importance for membrane-active helical AMPs. The effect on membrane interactions is, however, difficult to define, especially when relying on discrete experimental findings, considering that several other factors can affect its binding to, and insertion into, the membrane. Combining a number of different such observations with molecular dynamics simulations has helped us to better follow the binding process for kiadins, assessing differences in structuring, interactions with neutral or anionic liposomes and subsequent biological activity. Finding a common template for their mode of action, based on the correlation of experimental data with molecular modelling results, was, however, difficult (see Appendix A for a qualitative scoring of the peptides with respect to all these different parameters), and while it does provide an added insight into how linear helical AMPs work, as is unfortunately all too frequent with these peptides, it creates more questions than it answers.

Unlike experimental techniques, simulation studies are (i) hindered by model and method limitations and therefore often represent simplified conditions while (ii) usually sampling only one (this study) or, at the most, a limited number of peptides [62] instead of working with ample peptide populations (typically at micromolar concentrations). Furthermore, (iii) simulations have temporal and spatial scales that are far more limited than those used in experimental studies (currently ns–μs timescales compared to minutes or hours) due to limiting computational resources, (iv) which usually limits the observations to single, particular steps in the overall mode of action. For instance, the peptide needs to be positioned close to the membrane surface and already in a defined conformation, which is not what occurs in vitro in biophysical or biological experiments.

Nonetheless, we have found that data acquired using molecular modelling of kiadin/membrane interactions can be qualitatively compared to some experimental data to provide an increased insight into their mode of action in a manner that is relatively straightforward. This, however, requires taking into account the limits of both the experimental methods and MD simulations. For instance, CD signals include non-resolved contributions from different peptide populations in bulk solutions or are bound to membranes. SPR data provides information on membrane binding at a much later stage than MD simulations. For both CD and SPR experiments, the signal may be altered by processes such as conformational distortions or oligomerization upon membrane binding, which are not observable in MD simulations.

Experimental data from biological experiments have proven to be even more difficult to correlate with MD simulations, as there is an even larger gap between atomistic observations, i.e., even longer time scales, and the relatively high peptide concentrations required for both antibacterial activity and cytotoxicity toxicity measurements. However, as underlined in this and other studies [63,64,65], given the multifaceted effects that even a single positional mutation can have on the biological activity of AMPs, it is useful to find as many connections as possible with data from both biophysical and in silico experiments if we can hope to understand them.

This study provides some insight on how glycine substitutions affect the interactions of linear, helical AMPs with biological membranes and how this contributes to their modes of action and biological activities. However, it is difficult to come to solid conclusions based on a limited number of examples/peptides, and our work shows how even quite similar peptides (same size, charge and >50% sequence identity) can result in markedly different behaviours. To determine more generally applicable trends, an approach could be to test several different families of closely related peptides and then attempt to extract common behavioural aspects.

One encouraging outcome of our work is the observation, as can be discerned from Appendix A, that the antimicrobial potency, spectra and host cell cytotoxicity can be affected independently and connect with the differences observed at the atomistic level. This highlights that the combined use of in silico, biophysical and in vitro methods, albeit challenging, remains an unavoidable path in AMP research.

## Figures and Tables

**Figure 1 pharmaceutics-15-01433-f001:**
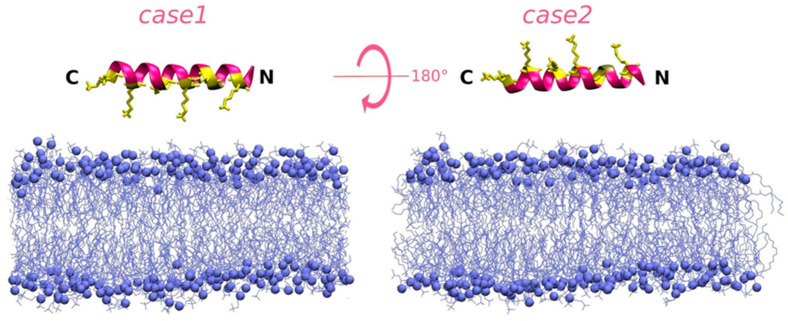
Representation of the initial conditions for the *case1* and *case2* simulation runs of *di*PGLa-H (KIAKVALKALKIAKVALKAL) interacting with the DLPC membrane taken at the 0 ns simulation time after an energy minimization procedure carried out on the peptide structure defined by the QUARK output [29,30]. Peptide molecules are shown as ribbon models with hydrophobic and Gly residue positions coloured magenta and polar ones coloured yellow. Lysine side chains are shown as yellow stick representations. DLPC membranes are shown in blue, with the phospholipid phosphate groups shown as spheres and acyl chains as lines.

**Figure 2 pharmaceutics-15-01433-f002:**
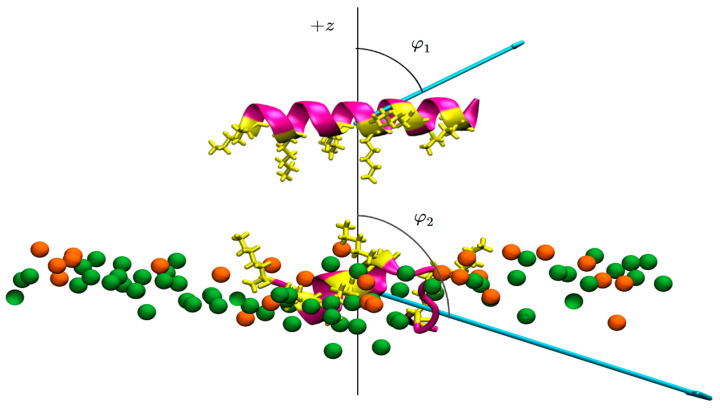
Peptide structure with the three-dimensional-hydrophobic moment, calculated using the 3D-HM server [48]. The upper part of the figure shows kiadin-2 above the membrane at 0 s (just after equilibration), with case1 orientation, and the lower part shows it after 300 ns interacting with the POPE:POPG membrane. Phospholipids are represented only by phosphate atoms (green and orange, respectively, for POPE and POPG). The 3D-hydrophobic moment vectors are represented by the cyan arrows, which lengths correspond to their calculated strengths (see Table 1), while the angles φ_1_ and φ_2_ between the normal membrane surface (+*z*-axis) and vectors define their direction. Representations for the other peptides are shown in the Appendix A).

**Figure 3 pharmaceutics-15-01433-f003:**
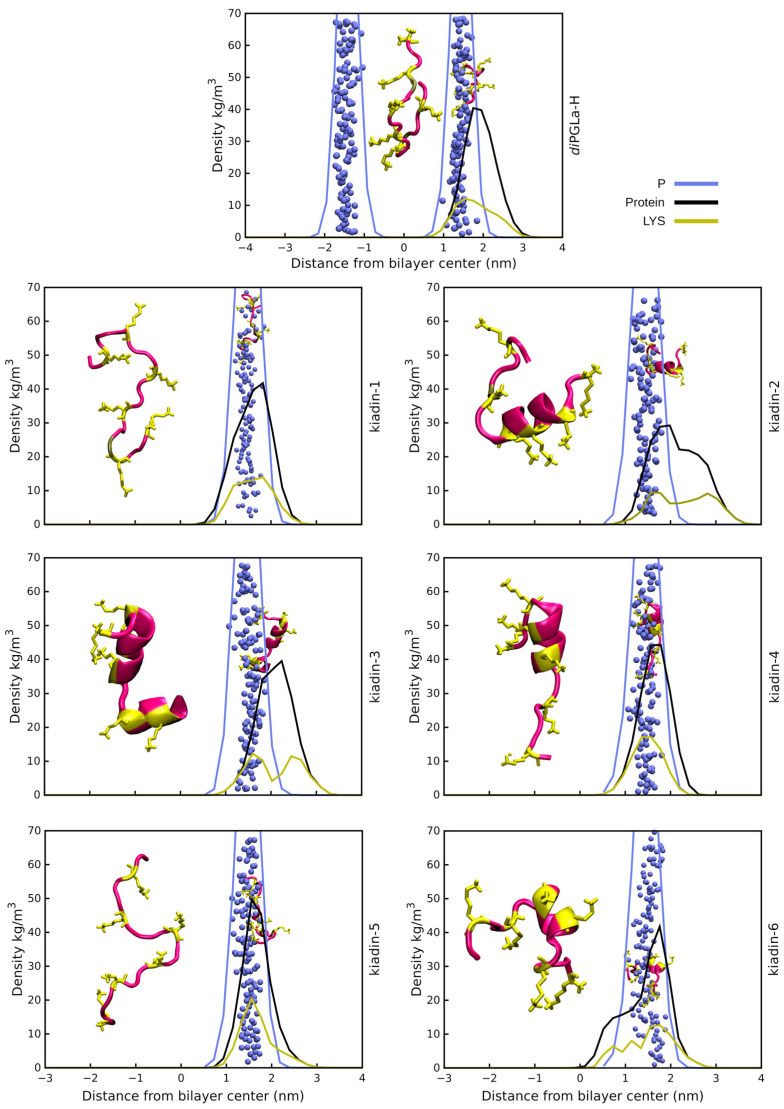
Density profiles for kiadin peptides with the DLPC membrane sampled over the last 100 ns of the simulation time (*case1*). The black curve shows the mass distribution of all atoms in the peptide and the blue curve the phosphorus atom mass distribution that represents the bilayer surface. The yellow plot curves instead show the distribution of the Lys residue side-chain atoms. The phosphorus atom distribution for both leaflets in the bilayer is shown only for *diPGLa*-H. At 300 ns (end of simulation), a snapshot was taken of the structure and is shown both at the membrane surface (smaller structure) and shifted to the side for clarity (larger structure).

**Figure 4 pharmaceutics-15-01433-f004:**
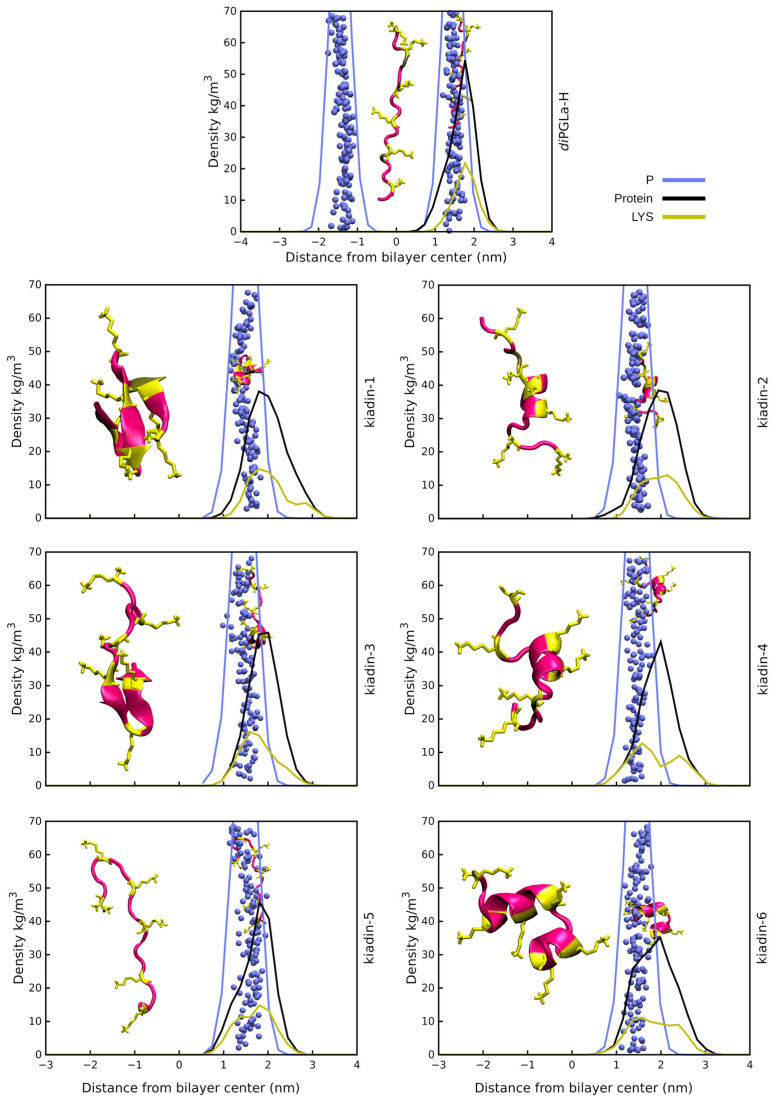
Density profiles for kiadin peptides with the DLPC membrane sampled over the last 100 ns of the simulation time (*case2*). The black curve shows the mass distribution of all atoms in the peptide and the blue curve the phosphorus atom mass distribution that represents the bilayer surface. The yellow plot curves instead show the distribution of the Lys residue side-chain atoms. The phosphorus atom distribution for both leaflets in the bilayer is shown only for *diPGLa*-H. At 300 ns (end of simulation), a snapshot was taken of the structure and is shown both at the membrane surface (smaller structure) and shifted to the side for clarity (larger structure).

**Figure 5 pharmaceutics-15-01433-f005:**
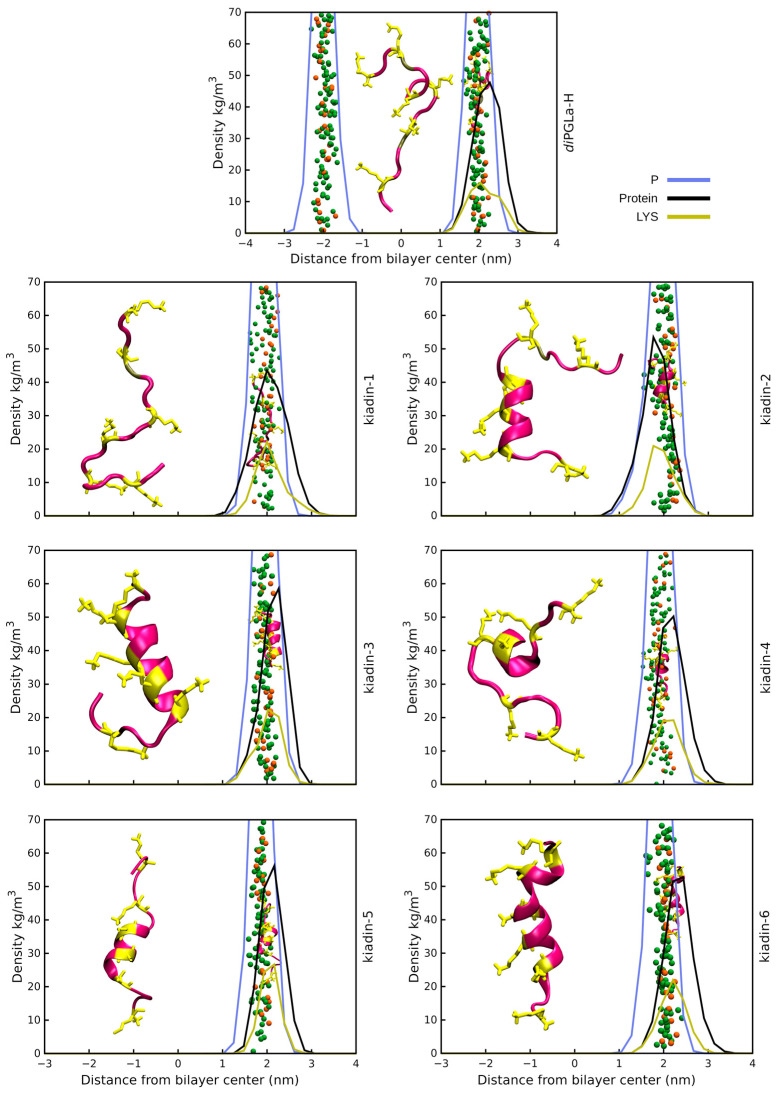
Density profiles for kiadin peptides with the POPE:POPG membrane sampled over the last 100 ns of the simulation time (*case1*). The black curve shows the mass distribution of all atoms in the peptide and the blue curve the phosphorus atom mass distribution that represents the bilayer surface. The yellow plot curves instead show the distribution of the Lys residue side-chain atoms. The phosphorus atom distribution for both leaflets in the bilayer is shown only for *diPGLa*-H. At 300 ns (end of simulation), a snapshot was taken of the structure and is shown both at the membrane surface (smaller structure) and shifted to the side for clarity (larger structure).

**Figure 6 pharmaceutics-15-01433-f006:**
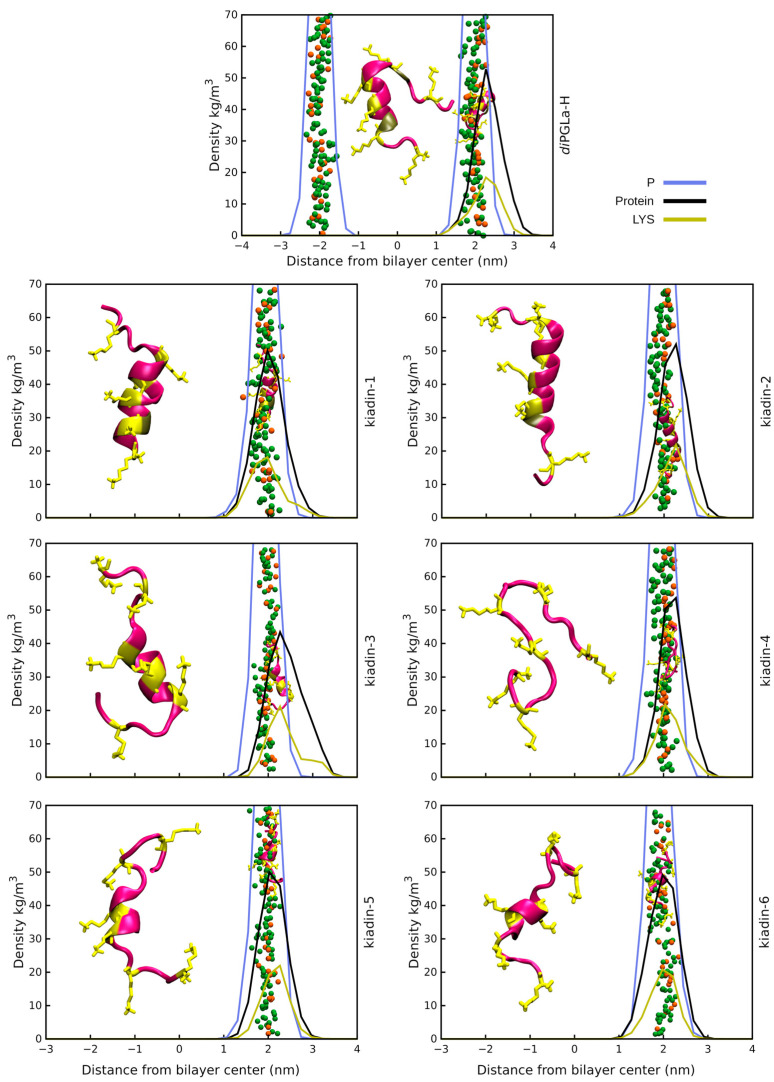
Density profiles for kiadin peptides with the POPE:POPG membrane sampled over the last 100 ns of the simulation time (*case2*). The black curve shows the mass distribution of all atoms in the peptide and the blue curve the phosphorus atom mass distribution that represents the bilayer surface. The yellow plot curves instead show the distribution of the Lys residue side-chain atoms. The phosphorus atom distribution for both leaflets in the bilayer is shown only for *diPGLa*-H. At 300 ns (end of simulation), a snapshot was taken of the structure and is shown both at the membrane surface (smaller structure) and shifted to the side for clarity (larger structure).

**Figure 7 pharmaceutics-15-01433-f007:**
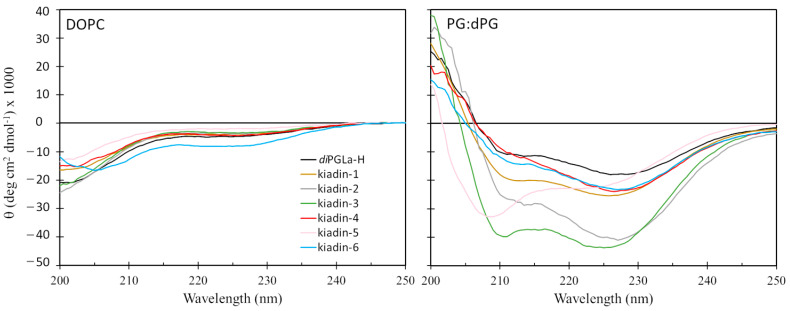
CD spectra of kiadins in the presence of neutral (DOPC) and anionic (PG:dPG) LUVs. Spectra are the accumulation of three scans carried out with 20 μM peptides.

**Figure 8 pharmaceutics-15-01433-f008:**
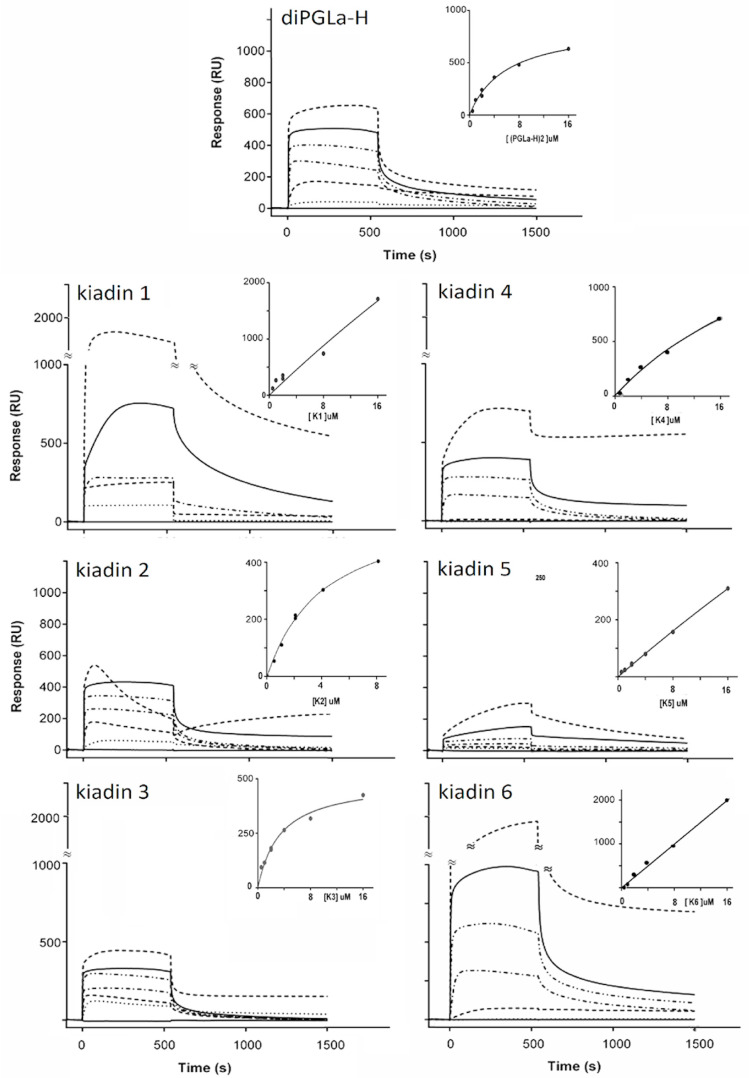
Surface plasmon resonance sensorgrams for binding of the indicated kiadin peptides to neutral DOPC LUVs and their respective binding curves (inset) determined by using the “affinity-steady state” mathematical model with Biacore BiaEvaluation software. Concentrations successively increased from 0.5 mM (⋅⋅⋅⋅⋅⋅⋅⋅⋅⋅⋅⋅⋅⋅) to 1 mM (------); 2 mM (-⋅-⋅-⋅-); 4 mM (-⋅⋅-⋅⋅-⋅⋅); 8 mM (⎯⎯⎯⎯) and 16 mM (------), washing with a buffer between each measurement.

**Figure 9 pharmaceutics-15-01433-f009:**
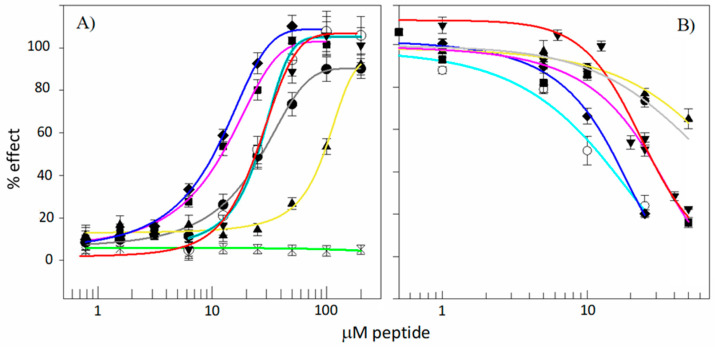
(**A**) Haemolytic effect and (**B**) effect on the viability of human MEC-1 lymphocytic cells. Haemolytic activity for kiadins was determined on a 0.5% erythrocyte suspension in PBS, while the MTT assay was carried out on 10^5^ cells/well in supplemented RPMI 1640 complete medium. *Di*PGLa-H (—○—); kiadin-1 (—▼—); kiadin-2 (—●—); kiadin-3 (—▲—); kiadin-4 (—■—); kiadin-5 (—🞵—); kiadin-6 (—◆—).

**Table 1 pharmaceutics-15-01433-t001:** Sequences and measured or estimated physicochemical characteristics of kiadins.

Peptide	Sequence ^a^	q ^b^	H ^c^	2D-HM ^d^	3D-HM ^e^ (*kTÅ/e*)				
					Simulation Time 0 ns	Simulation Time 300 ns			
					H_2_O	DPLC		PE:PG	
						*case1*	*case2*	*case1*	*case2*
*di*PGLa-H	KIAKVALKALKIAKVALKAL-NH_2_	+7	0.44	0.42	23.5 ± 0.5	13.7	68.7	42.7	41.1
kiadin-1	KIAKVALKALKIAK**G**ALKAL-NH_2_	+7	0.38	0.48	18.1 ± 0.7	27.4	18.1	12.4	69.5
kiadin-2	KIAK**G**ALKALKIAKVALKAL-NH_2_	+7	0.38	0.41	23.2 ± 0.9	37.0	52.7	46.2	67.1
kiadin-3	KIAK**G**ALKALKIAK**G**ALKAL-NH_2_	+7	0.32	0.46	23.9 ± 0.9	38.2	37.9	44.6	39.8
kiadin-4	KI**G**KAL**G**KALKAL**G**KAL**G**KA-NH_2_	+7	0.21	0.65	20.2 ± 0.4	47.8	34.9	27.8	25.5
kiadin-5	KIA**G**KA**G**KIAKIA**G**KA**G**KIA-NH_2_	+7	0.16	0.37	16.3 ± 0.8	20.7	22.0	64.3	27.1
kiadin-6	KIALKALKALKAL**G**KALKAL-NH_2_	+7	0.40	0.49	20.2 ± 0.7	26.0	20.3	65.2	49.9

^a^ Conserved residues with respect to *di*PGLa-H are in orange. Residue differences deriving from the different design methods are in black, and Gly substitutions in bold; ^b^ charges are defined at pH = 7; ^c^ mean per residue hydrophobicity (as determined using HeliQuest); ^d^ mean per residue 2D-hydrophobic moment (amphipathicity, as determined using HeliQuest); ^e^ 3D-hydrophobic moment calculated using the 3D-HM tool for the peptide at 0 ns, after the equilibration and averaged over 4 cases, and after a 300 ns simulation time near DLPC or POPE:POPG (PE:PG) membranes; *case1* = polar side initially facing the membrane surface and *case2* = hydrophobic side initially facing membrane surface. In (kTÅ/e), k is Boltzmann’s constant, T is the temperature in K, Å is 10^−10^ m and e is the electron charge.

**Table 2 pharmaceutics-15-01433-t002:** Antibacterial activity and toxicity of *di*PGLa-H and kiadin peptides.

Peptide	*E. coli*		*S. aureus*		RBC	MEC-1
	MIC ^a^	MBC ^b^	MIC	MBC	HC_50_ ^c^	IC_50_ ^d^
*di*PGLa-H	1.5	1.5	0.75	1.5	22 ± 2	9 ± 3
kiadin-1	0.75	0.75	0.5–1	1.5	22 ± 2	20 ± 3
kiadin-2	0.25–0.5	0.5–1	8–16	8–16	26 ± 6	>50
kiadin-3	4	16	16–32	32	95 ± 10	>50
kiadin-4	8	16	16–32	32	11 ± 1	13 ± 1
kiadin-5	>64	>64	>64	>64	n.d	n.d.
kiadin-6	1–2	2–4	4	4	10.5 ± 0.5	14 ± 1

^a^ Minimal inhibitory concentration—the lowest concentration of the peptide showing no visual bacterial growth after an overnight incubation; ^b^ minimal bactericidal concentration—the lowest concentration with no visible bacterial growth on MH agar after an overnight incubation of peptide–bacteria aliquots taken from wells corresponding to the MIC, 2 × MIC and 4 × MIC; ^c^ HC_50_ value was calculated as the concentration causing 50% haemolysis; ^d^ IC_50_ value was estimated as the concentration of peptides causing a 50% decrease in viability, calculated using GraphPad Prism software. All values are expressed in mM concentrations.

## Data Availability

The data presented in this study are available in the Appendix A.

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
