# Peer review of "Relating Molecular Dynamics Simulations to Functional Activity for Gly-Rich Membranolytic Helical Kiadin Peptides"

_pharmaceutics, 2023, doi:10.3390/pharmaceutics15051433_

Round 1

Reviewer 1 Report

The manuscript by Tomislav Rončević et al. describe the properties of seven kiadin antimicrobial peptides. Authors strive to find a correlation between MD analyses and results from in vitro assays.

Well recognizing the value of these type of studies the overall conclusions of the manuscript suffer of the limited number of examples/peptides taken into consideration, and that should be addressed in the conclusions

The manuscript has to be revised before publishing. Firstly, author should make an effort to reduce the length of the manuscript and avoid redundancies. As an example, paragraph 3.2 has to be compacted in one single bullet list. Besides, discussion and conclusions have to be shortened and too general or obvious considerations should be avoided. The section of MM has to become a succinct description of the methods applied and state-of-art considerations or explanations of the selected methods have to be reported in the results or figure legends.

Figures 7 and 8 have to be improved in quality, particularly the different colors or different line styles have to more clearly different.

In paragraph 3.3, there’s an important concept that should be treated in a more extensive discussion (lines 165-166 of page 18).

In the Discussion (line 349-355 of page 25) authors should rephrase the paragraph, in a way that the observation are not to be considered rule, since the little number of observations cannot allow to define general rules.

The title should reflect the fact that 50% of the manuscript is on MD. A frank concise list of the value of the use of MD to complement direct measures in vitro should also be addressed.

Author Response

We thank the reviewers for their comments. We carefully considered all inputs and revised the manuscript by incorporating reviewers' suggestions.  

The manuscript by Tomislav Rončević et al. describe the properties of seven kiadin antimicrobial peptides. Authors strive to find a correlation between MD analyses and results from in vitro assays. Well recognizing the value of these type of studies the overall conclusions of the manuscript suffer of the limited number of examples/peptides taken into consideration, and that should be addressed in the conclusions

We agree with the referee that one cannot come to very solid conclusions based on a limited number of examples/peptides, however our work has been useful to point out how even quite similar peptides (same size, charge and >50% sequence identity) can result in markedly different behavior, so that we feel that the approach should be to test several different closely related peptide families and then try to extract any common trends. This point has been made in the conclusions section.

The manuscript has to be revised before publishing. Firstly, author should make an effort to reduce the length of the manuscript and avoid redundancies. As an example, paragraph 3.2 has to be compacted in one single bullet list. Besides, discussion and conclusions have to be shortened and too general or obvious considerations should be avoided. The section of MM has to become a succinct description of the methods applied and state-of-art considerations or explanations of the selected methods have to be reported in the results or figure legends.

Section MM has been shortened, especially the part related to the MD methodology and redundancies have been removed. Information on equilibration details and some analysis details have been placed in the supplement. Paragraph 3.2 has been changed according to the suggestion. 

Figures 7 and 8 have to be improved in quality, particularly the different colors or different line styles have to more clearly different.

The improved figures with the higher resolution have been placed in the manuscript. Colouring in Figure 7 has been changed.

In paragraph 3.3, there’s an important concept that should be treated in a more extensive discussion (lines 165-166 of page 18).

This concept has been better explained in the results section and is also commented in the discussion section.

In the Discussion (line 349-355 of page 25) authors should rephrase the paragraph, in a way that the observation are not to be considered rule, since the little number of observations cannot allow to define general rules.

This part has been amended and toned down not to appear as a rule, but as an indication as to how to interpret CD behaviour.

The title should reflect the fact that 50% of the manuscript is on MD. A frank concise list of the value of the use of MD to complement direct measures in vitro should also be addressed.

The title has been changed to better reflect the manuscript content.

Reviewer 2 Report

This paper addresses the role of glycine substitution in a group of antimicrobial peptides called kiadins. It is a follow-up study to reference 23, titled "Antibacterial Activity Affected by the Conformational Flexibility in Glycine-Lysine-Based Alpha-Helical Antimicrobial Peptides," published in the Journal of Medicinal Chemistry in 2018.

Some of the information in this manuscript has already been described in the former paper and should be clearly stated as not new. For example, circular dichroism in PG:dPG and antimicrobial activities in E. coli and S. aureus are identical to the former reference. The authors should explicitly state that this information was taken from other published studies. Although the results for hemolytic activity and cytotoxicity are not identical, they are similar to the values reported in reference 23.

In the molecular dynamics simulation, the authors used the structure of peptides predicted by the QUARK server, in an alpha-helical configuration. However, as previously published for these molecules, peptides in buffer (absence of membranes) are typically in a random conformation and only acquire structure upon binding to membranes. MD simulations show that some structures, such as kiadin 4, are less stable in water. Can the authors comment on how this affects the MD simulations? Can this conformational switch upon membrane binding be modeled by MD? Would this be a limitation in MD simulations that could explain inconsistencies between simulations and experimental results?

The newest information in this paper is the molecular dynamics simulation of the peptides and how it correlates with experimental data. As the authors note in their paper, finding correlations is difficult because different initial conditions in MD simulations, such as the orientation of the peptide relative to the membrane, can produce different results. In this context, it would be helpful to include a summary table with all the results presented in a qualitative manner (e.g., using +, ++, and +++ instead of actual values). The authors could relate the predicted structure in MD with the CD results, the binding affinities detected by surface plasmon resonance, and the measured antimicrobial activity. This would help readers to put all the results in context.

The authors performed surface plasmon resonance on peptides binding to DOPC liposomes. In this case, as indicated by CD spectroscopy, the peptides do not structure upon binding. However, in anionic LUVs (PG:dPG), the peptides do acquire structure upon binding. Measuring the binding of the peptides to anionic LUVs would be interesting to compare the results with neutral vesicles and could provide more insight to explain the similarities/differences between simulations and experimental results.

The title should focus more on the experimental setup designed by the authors. "Influence of Glycine Substitutions on the Mode of Action of Membranolytic Helical Antimicrobial Peptides" is too general. The authors could not find specific rules for the influence of glycine substitution, and even if they did, these rules would not generalize to all AMPs, only to kiadins. A more appropriate title could be "Influence of Glycine Substitutions in Kiadin Antimicrobial Peptides."

Minor comments:

Please clarify in the methods section that the peptides were not designed in this study and that sequences come from reference 23.

I appreciate the HPLC chromatograms provided for the peptides. I suggest the authors also provide the mass spectrometry spectra.

Please specify the peptide concentration in the CD spectra, as well as the cuvette width and the recorded wavelength range.

Please specify the name of the strains used for antimicrobial assays, and indicate if they are commercially available or clinical isolates.

What is the origin of red blood cells? Are they commercially available?

What is the reasoning behind utilizing MEC-1 cells for cytotoxicity assessments? Fibroblast or hepatocyte cell types are frequently employed, but the authors decided to use a cell line derived from B-chronic lymphocytic leukemia.

Author Response

We thank the reviewers for their comments. We carefully considered all inputs and revised the manuscript by incorporating reviewers suggestions.  

This paper addresses the role of glycine substitution in a group of antimicrobial peptides called kiadins. It is a follow-up study to reference 23, titled "Antibacterial Activity Affected by the Conformational Flexibility in Glycine-Lysine-Based Alpha-Helical Antimicrobial Peptides," published in the Journal of Medicinal Chemistry in 2018. Some of the information in this manuscript has already been described in the former paper and should be clearly stated as not new. For example, circular dichroism in PG:dPG and antimicrobial activities in E. coli and S. aureus are identical to the former reference. The authors should explicitly state that this information was taken from other published studies

Hemolytic activity and antimicrobial activity were published in ref 23, and this is indicated in the manuscript in the relevant sections, however cytotoxic activity on MEC-1 cells is new and all the activities are now gathered in a new context. We have made it more explicit that some data was previously published.

In the molecular dynamics simulation, the authors used the structure of peptides predicted by the QUARK server, in an alpha-helical configuration. However, as previously published for these molecules, peptides in buffer (absence of membranes) are typically in a random conformation and only acquire structure upon binding to membranes. MD simulations show that some structures, such as kiadin 4, are less stable in water. Can the authors comment on how this affects the MD simulations? Can this conformational switch upon membrane binding be modelled by MD? Would this be a limitation in MD simulations that could explain inconsistencies between simulations and experimental results?

We agree with the referee that helical AMPs are normally not structured in bulk solution. Unfortunately,  currently available modelling resources make it difficult to simulate the complete folding from a random coil to helical structure on membrane interaction, within reasonable simulation times. Starting with a defined conformation is a currently accepted method for simulating the interaction within these constraints, as  a useful compromise, and we have attempted to increase its validity by using two different approach orientations. In effect, in this type of  simulation the peptide unfolds from the helix but approaches the membrane at the same time, so it may find itself in a partly folded form at the membrane surface. This procedure is our best approach in an attempt to come as close as possible to a structure of partially folded peptide in the membrane as measured by, for example, CD spectra.

The referee is right to point out that kiadins have different stabilities in water observed in simulations and therefore approach with a more or less preserved helical structuring towards the membrane surface. This adds to their diversity in the binding process. But it is also true that the initial configuration affects the simulation results (as mentioned for example in ref. BBA - Biomembranes 1861 (2019) 1409–1419) and can be one of the limitations that can lead to inconsistencies with experiments. However, simulation time and insufficient simulation cases to properly represent the peptide fluctuations are more likely factors that limit simulations in well representing real systems.

The newest information in this paper is the molecular dynamics simulation of the peptides and how it correlates with experimental data. As the authors note in their paper, finding correlations is difficult because different initial conditions in MD simulations, such as the orientation of the peptide relative to the membrane, can produce different results. In this context, it would be helpful to include a summary table with all the results presented in a qualitative manner (e.g., using +, ++, and +++ instead of actual values). The authors could relate the predicted structure in MD with the CD results, the binding affinities detected by surface plasmon resonance, and the measured antimicrobial activity. This would help readers to put all the results in context.

We thank the referee for this suggestion. We have created a table in which the results were assessed in a qualitative manner as indicated, and furthermore also used a heat-map type colour scheme to make it visually easier to assess. It does allow to appreciate the different behaviour of the kiadins and emphasise how antimicrobial potency can to some extent be affected independently. We however feel it was not justified to place and discuss it in the manuscript, but rather ain the supplementary material and briefly comment it in the manuscript.

The authors performed surface plasmon resonance on peptides binding to DOPC liposomes. In this case, as indicated by CD spectroscopy, the peptides do not structure upon binding. However, in anionic LUVs (PG:dPG), the peptides do acquire structure upon binding. Measuring the binding of the peptides to anionic LUVs would be interesting to compare the results with neutral vesicles and could provide more insight to explain the similarities/differences between simulations and experimental results.

We agree that an SPR study of interaction with anionic LUV can be interesting, but our reason for studying neutral LUVs was to supplement CD data which showed little apparent interaction for most kiadins but significantly varying cytotoxicities. The SPR in effect allowed us to discern quite different binding behaviour. With anionic LUVs, electrostatic interactions tend to dominate, which can make this more difficult.

The title should focus more on the experimental setup designed by the authors. "Influence of Glycine Substitutions on the Mode of Action of Membranolytic Helical Antimicrobial Peptides" is too general.  The authors could not find specific rules for the influence of glycine substitution, and even if they did, these rules would not generalize to all AMPs, only to kiadins. A more appropriate title could be "Influence of Glycine Substitutions in Kiadin Antimicrobial Peptides."

 The title has been changed to better reflect the manuscript content and use of Kiadins, and in compliance to rhe referee 1 also makes reference to the MD method used. 

Minor comments:

Please clarify in the methods section that the peptides were not designed in this study and that sequences come from reference 23.

This is made clear in both the materials and methods section and in the results.

I appreciate the HPLC chromatograms provided for the peptides. I suggest the authors also provide the mass spectrometry spectra.

The MS spectra have been included in the supplementary materials as Figure S1B.

Please specify the peptide concentration in the CD spectra, as well as the cuvette width and the recorded wavelength range.

This has been done.

Please specify the name of the strains used for antimicrobial assays, and indicate if they are commercially available or clinical isolates.

Microbiological assays were performed with commercially available strains, and in particular we used: E. coli ATCC25922 and S aureus ATCC 25923. This was specified in the results section 3.5 (Antibacterial activity). We have now placed this information in the M&M, we thank the reviewer for pointing this out.

What is the origin of red blood cells? Are they commercially available?

The RBC were kindly donated by a healthy donor who gave informed consent (one of the authors) and observed the ethical principles of the Helsinki declaration. The MEC-1 cell line was kindly donated by prof. P. Macor and this has been indicated in the M&M with the appropriate references.

What is the reasoning behind utilizing MEC-1 cells for cytotoxicity assessments? Fibroblast or hepatocyte cell types are frequently employed, but the authors decided to use a cell line derived from B-chronic lymphocytic leukemia.

We agree with the reviewer comment that fibroblasts and hepatocyte cell lines are frequently used for screening assays, they are robust and grow in adhesion. However, for our purpose we have chosen to perform cytotoxic assays on cells growing in suspension having all the membrane surface accessible to peptides in solution, moreover their proliferative activity render them more sensitive to any toxic effect. MEC-1 cells resemble circulating cells (as with erythrocytes) useful to highlight a potential toxic effect at a systemic level. This model is widely used even for flow cytometry assays (not in this manuscript), since their membrane remains undamaged throughout all experimental procedures.

Round 2

Reviewer 2 Report

The authors have addressed all my comments.